# Two User-Friendly Molecular Markers Developed for the Identification of Hybrid Lethality Genes in *Brassica oleracea*

**Zhiliang Xiao, Congcong Kong, Fengqing Han, Limei Yang, Mu Zhuang, Yangyong Zhang, Yong Wang, Jialei Ji, Zhansheng Li** 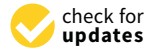**, Zhiyuan Fang \* and Honghao Lv \***

Institute of Vegetables and Flowers, Chinese Academy of Agricultural Sciences, No. 12 Zhongguancun South Street, Beijing 100081, China; 13051379638@163.com (Z.X.); kongcongcong@caas.cn (C.K.); hanfengqing@caas.cn (F.H.); yanglimei@caas.cn (L.Y.); zhuangmu@caas.cn (M.Z.); zhangyangyong@163.com (Y.Z.); wangyong@163.com (Y.W.); jijialei@caas.cn (J.J.); lizhansheng@caas.cn (Z.L.)
\* Correspondence: fangzhiyuan@caas.cn (Z.F.); lvhonghao@caas.cn (H.L.); Tel.: +010-82108756 (H.L.)

**Abstract:** Cabbage (*Brassica oleracea*) is an important vegetable crop that is cultivated worldwide. Previously, we reported the identification of two dominant complementary hybrid lethality (HL) genes in cabbage that could result in the death of hybrids. To avoid such losses in the breeding process, we attempted to develop molecular markers to identify HL lines. Among 54 previous mapping markers closely linked to *BoHL1* or *BoHL2*, only six markers for *BoHL2* were available in eight cabbage lines (two *BoHL1* lines; three *BoHL2* lines; three lines without *BoHL*); however, they were neither universal nor user-friendly in more inbred lines. To develop more accurate markers, these cabbage lines were resequenced at an ~20× depth to obtain more nucleotide variations in the mapping regions. Then, an InDel in *BoHL1* and a single-nucleotide polymorphism (SNP) in *BoHL2* were identified, and the corresponding InDel marker MBoHL1 and the competitive allele-specific PCR (KASP) marker KBoHL2 were developed and showed 100% accuracy in eight inbred lines. Moreover, we identified 138 cabbage lines using the two markers, among which one inbred line carried *BoHL1* and 11 inbred lines carried *BoHL2*. All of the lethal line genotypes obtained with the two markers matched the phenotype. Two markers were highly reliable for the rapid identification of HL genes in cabbage.

**Keywords:** hybrid lethality; Brassica oleracea; breeding; InDel marker; KASP marker



## 1. Introduction

Cabbage (*Brassica oleracea* L. var. *capitata*), a cole crop species, is a vegetable of worldwide economic importance due to its strong resistance, wide adaptability, favorable taste and health-related value [1,2]. Currently, compared with traditional varieties, most commercial cabbage cultivars are first-generation hybrids with strong heterosis (hybrid vigor) that are widely used in cabbage production [3,4].

Heterosis is defined as the phenomenon that the progeny of diverse varieties of a species or crosses between species exhibit greater biomass, development speeds, and fertility than both parents. The greater the differences in the genetic relationships of the parents, the greater the advantages of the relative traits that can potentially complement each other. Heterosis is used to improve yield, uniformity, and vigor and is exploited in the breeding of corn, sorghum, rice, sugar beet, onion, spinach, sunflower, broccoli, and hemp, among others [5–10].

Hybrids do not necessarily exhibit superiority or vigor but sometimes show weakness or even death, resulting in unexpected losses during the breeding process. Hybrid weakness or lethality has been reported in important crops, such as Arabidopsis, lettuce, rice, ginseng, bean, mimulus, cotton, potato, and tomato [11–18]. Additionally, in 2016, we reported the identification of hybrid lethality (HL) in cabbage that hinders the exploitation of heterosis and causes economic losses [19]. HL is a type of reproductive segregation that

occurs after mating or fertilization and causes F1 hybrid death. Incompatibility of parental genes or chromosomes result in decreases in the viability and fertility of interspecific hybrids [20,21]. Incompatibilities in internal genetic factors cause hybrid weakness/necrosis, breakdown, sterility, and unviability. In other words, HL is a type of hybrid weakness, in contrast to heterosis.

Molecular marker technology has been widely used as a valuable tool in molecular biology research, including the analysis of genetic diversity, prediction of heterosis, identification of germplasms, gene mapping, and marker-assisted selection breeding [22–24]. There are dozens of different molecular markers, such as random amplified polymorphic DNA (RAPD), sequence-characterized amplified region (SCAR), simple sequence repeat (SSR), insertion-deletion (InDel) and competitive allele-specific PCR (KASP) markers. According to the methods for detecting DNA polymorphisms, molecular markers can be divided into four categories: (1) fragment length polymorphisms (RFLPs), whose detection is based on Southern hybridization; (2) RAPD, SSR, sequence-related amplified polymorphism (SRAP), and inter simple sequence repeat (ISSR) markers, detected based on polymerase chain reaction (PCR); (3) amplified fragment length polymorphisms (AFLPs) and Cardiac Arrhythmia Pilot Study (CAPS) markers, detected based on a combination of PCR and restriction enzyme digestion technology; and (4) the recent ones: InDel, single-nucleotide polymorphism (SNP), and KASP molecular markers. Molecular marker technology provides one of the most important auxiliary methods for genetic breeding and represents a powerful auxiliary approach for selecting and cultivating new cabbage cultivars that possess multiple superior traits. A number of molecular markers for cabbage have been developed and successfully applied for cabbage breeding, such as the SSR marker Frg13 for cabbage Fusarium wilt resistance [25,26], the InDel markers BoID0709 and BoID0992 associated with self-compatibility [27], the *Rfo* marker BnRFO [28,29], and the KASP marker K13 and the SCAR marker ST11 900 for dominant genic male sterility [30,31].

We have previously reported the identification of HL in cabbage [19]. Genetic analyses revealed that HL in cabbage was controlled by two complementary dominant genes, *BoHL1* and *BoHL2*, which were then mapped to chromosomes C1 and C4, respectively [32]. In hybrid breeding, lethal cabbage hybrid combinations cannot be predicted, and their appearance inevitably causes economic losses. Therefore, it is necessary to identify materials carrying HL genes by using high-efficiency and user-friendly diagnostic molecular markers. In the current study, we developed molecular markers based on the whole-genome resequencing data of a series of inbred lines to identify cabbage lines carrying HL genes, which were further validated in 138 accessions to validate their reliability.

## 2. Materials and Methods

### 2.1. Plant Materials

Eight cabbage lines were used in our study: two *BoHL1* lines (09-211 and 11-204) and three *BoHL2* lines (09-222, 10-260 and 11-176), in which the hybrid plants of the *BoHL1* lines and *BoHL2* lines exhibited 100% seedling mortality at a late stage of growth, and three control lines (87-534, 96-100 and 01-20) without the HL genes. Additionally, 138 Brassica crop species (106 cabbage lines, 10 broccoli lines, 11 Chinese cabbage lines and 11 Chinese kale lines) were identified and used to validate the specific markers. All plant materials were provided by the Cabbage and Broccoli Research Group, Institute of Vegetables and Flowers, Chinese Academy of Agricultural Sciences (IVF-CAAS).

### 2.2. Primer Design

First, we attempted to identify user-friendly markers from the fine mapping markers of *BoHL1* and *BoHL2* in the eight cabbage lines. We identified one SSR marker and 28 InDel markers for *BoHL1* and 27 InDel markers for *BoHL2* that have been used for the fine mapping of HL genes [19,32]. Furthermore, all eight cabbage lines were resequenced to an ~20× depth to obtain additional nucleotide variations, which were deposited in the NCBI Sequence Read Archive (SRA) under BioSample accessions (SAMN06841129-

30, SAMN17385836-47). For *BoHL1*, we searched the genomic region in the *BoHL1* fine mapping region for nucleotide variations (SNPs, InDels and structural variants (SVs)) between the *BoHL1* lines and no-*BoHL1* lines (*BoHL2* lines and three control lines); using the same strategy, we searched nucleotide variations between the *BoHL2* lines and no-*BoHL2* lines (*BoHL1* lines and three control lines) from the fine mapping region of *BoHL2*. InDels ≥3 bp long and SVs were used to design InDel markers; InDels ≤2 bp long and SNPs were selected to design KASP markers following the PolyMarker pipeline). The primers were designed with Premier 5 (Premier Biosoft International, Palo Alto, CA, USA). Nucleotide variations associated with the markers were verified by Sanger sequencing, and sequence alignment was conducted with DNAman.

### 2.3. Genotyping

The genomic DNA of all cabbage lines was extracted from young leaves according to the modified CTAB protocol. The concentration of DNA was determined using an ND-1000 system (NanoDrop Technologies, Inc. Wilmington, DE, USA), and the DNA was then diluted to 40–50 ng/μL. For the InDel markers, PCR was performed using 2× Taq Master Mix (Vazyme, Nanjing, China). The reaction system and program followed the manufacturer's instructions. The PCR-amplified products were checked by 1% agarose gel electrophoresis (150 V) and subjected to sequencing. The KASP assay was performed following the protocol of LGC Genomics (Berlin, Germany), as reported previously [33]. An Applied Biosystems Viia 7 real-time PCR system (Applied Biosystems, Foster City, CA, USA) was used to detect allele-specific fluorescence, and the 122 genotypes were called with Viia 7 software, v1.0. To analyze the phenotypes and genotypes, *BoHL1* genotypes and normal genotypes were recorded as "aa" and "bb", respectively; *BoHL2* genotypes and normal genotypes were recorded as "cc" and "dd", respectively; and heterozygous genotypes and other genotypes were recorded as "h" and "ee", respectively.

### 2.4. Phenotyping

The cabbage lines identified by using the molecular markers were crossed with 09-211 or 09-222. The $F_1$ seeds will be sown in the field and observed the growth of the seedlings. Lethal symptoms in cabbage performance, e.g., retarded growth, wilting, and chlorosis, gradually appeared at approximately 30 days after germination. The positive control HL cross combination was 09-211 × 09-222, and the negative control cross combination was 09-211 × 87-534.

## 3. Results

### 3.1. Identification of Linkage Markers from the Fine-Mapping Primers

First, we identified the primers used for the fine mapping of HL genes in the eight cabbage lines, and the results are shown in Table 1. Among the markers linked to the *BoHL1* gene, none of the twenty-nine markers showed 100% accuracy in the eight cabbage lines, and eight markers presented 87.5% accuracy. These eight markers all showed genotypes that did not match the phenotype of cabbage lines 11-204. The identification of 50% false positives in two *BoHL1* lines indicated that these could not be used for identifying HL. Among the markers linked to *BoHL2*, six InDel markers, HL205, HL230, L27, L237, L43 and L411, showed 100% accuracy in distinguishing the eight cabbage lines (Figure 1), and six markers showed 87.5% accuracy (HL202, HL204, HL207, HL208, HL209 and HL235). Then, four InDel markers (HL230, L27, HL209 and L237) were used to identify 138 inbred lines, and the results are shown in Table S4. We found that 89.8% (124/138) of the inbred lines showed consistent genotypes at the four InDel markers, suggesting that the four markers presented relatively consistent evaluation results. Among these cabbage lines, 13 showed the same genotypes as the *BoHL2* lines, including G4, G11, G18, G29, G43, G73, G81, G87, G90, G91, G92, G93 and G95, suggesting that these cabbage lines might carry the *BoHL1* gene. However, some of the inbred lines showed different genotypes for HL230, L27, HL209 and L237, such as G67 (dd; dd; h; dd), P4 (e; dd; dd; dd), J7 (dd; dd; e; dd), QH11

(dd; dd; dd; e) and BC6 (e; dd; dd; dd). Moreover, some markers exhibited nonspecific amplification in certain lines, showing no amplified bands (G46, G70 for HL230; G62 for L237) or different bands (G70, G75, J7 and Y2 for HL209; P4, BC6 for HL230). These results indicated that these markers are universal in inbred lines with different backgrounds and that more user-friendly molecular markers for HL genes should be developed.

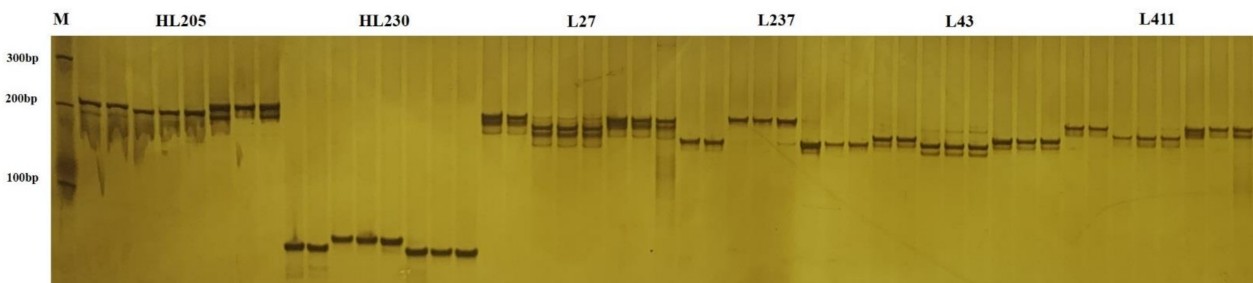

**Figure 1.** Detection of eight cabbage lines by using six *BoHL2* markers. Six InDel markers (HL205, HL230, L27, L237, L43 and L411) showed 100% accuracy in distinguishing the eight cabbage lines.

### 3.2. Development of the InDel Marker MBoHL1 Associated with BoHL1

The resequencing data of the eight cabbage lines were mapped to the TO1000 cabbage reference genome (http://plants.ensembl.org/Brassica_oleracea/Info/Index (accessed on 20 September 2020)). According to our previous study, *BoHL1* and *BoHL2* were located on chromosome C1 (42.79–42.90 Mb) and chromosome C4 (46.33–46.40 Mb), respectively [32]. In the *BoHL1* mapping location, seven nucleotide variations (4 SNPs, 2 InDels and 1 SV) were identified in *BoHL1* lines in comparison with the no-*BoHL1* lines (Table S1). The SV insert (CCCATTATGGCCGGTCCTGGGCAAAGCCTAACGAAACATTGGCGTTAGGCCTCCAAAAATTTTGGAAAAAATTACATAGAAAAAAGCTCCCAAAATTTGTTTTTTAG GCCCCTAAATTTACAAAAAATTTTTTAGCTGAAATTTTTTAAAAACCGCCTGCAGCCCCCTAAATCTCAGGGCCGGCCCCGAGTTAC) was confirmed by Sanger sequencing, and then the InDel marker MBoHL1 was developed to amplify the sequence. The MBoHL1 sequences were MBoHL1F (5′ to 3′): TAAACAAGGCTGGATAACAATAC and MBoHL1R (5′ to 3′): TCACCGGTCTGCTTCTTGAC. The graph of the agar gel results showed that the lanes containing the amplification products of the *BoHL1* lines were approximately 200 bp longer than the lanes of the other lines, and the alignment results also suggested that the SV carrying the 195-bp insert was only present in the *BoHL1* lines (Figure 2). Then, the InDel marker MBoHL1, which showed 100% accuracy in the eight lines, was developed and applied to identify whether more inbred lines carried *BoHL1*.

### 3.3. Development of the KASP Marker KBoHL2 Associated with BoHL2

No common nucleotide variations were identified in the *BoHL2* lines in the region containing *BoHL2* (46.33–46.40 Mb). Hence, we expanded the search for mutations to 50 kb up- and downstream of the *BoHL2* region, and two SNPs and two InDels were identified in the BoHL2 lines in comparison with the no-BoHL2 lines (Table S2). We chose two SNP mutations: SNP1, a T to C in 46,333,244, and SNP2, a G to A in 46,412,513. Two primers were designed for confirmation by Sanger sequencing. However, the sequencing results produced incorrectness for SNP2: the G to A mutation at 11-176 was not present. The sequencing results for SNP1 (46333244) confirmed its accuracy in the eight lines, and the KASP marker KBoHL2 was then designed. The KASP primer sequences were Primer_Allele X (5′ to 3′): CTCCGGGTTTTCCCGATATCC; Primer_Allele Y (5′ to 3′): TACTCCGGGTTTTCCCGATATCT; Primer_Common (5′ to 3′): GCCTGGACGATGGTAGCTCGAA. We evaluated the eight cabbage lines to assess the applicability and accuracy of KBoHL2, and genotyping with KBoHL2 showed 100% accuracy in the eight lines (T/T allele for three *BoHL2* lines and C/C allele for five no-*BoHL2* lines) (Figure 3), indicating that KBoHL2 was a useful marker for the *BoHL2* gene.

**Table 1.** Phenotyping of 54 pairs of fine mapping markers in eight cabbage lines. *BoHL1* genotypes, normal genotypes, heterozygous genotypes, other genotypes and no amplified bands were recorded as "aa", "bb", "h", "cc" and "no", respectively.

| Phenotyping | Primers | Physical Location | *BoHL1* Lines | | *BoHL2* Lines | | | Control Lines | | |
|---|---|---|---|---|---|---|---|---|---|---|
| | | | 09-211 | 11-204 | 09-222 | 10-260 | 11-176 | 87-534 | 96-100 | 01-20 |
| *BoHL1* | LTSSR44 | C1: 23318181 | aa | aa | u | bb | bb | u | bb | bb |
| | HL001 | C1: 39731894 | aa | h | bb | bb | bb | bb | bb | bb |
| | HL012 | C1: 39968608 | aa | h | aa | bb | aa | bb | bb | bb |
| | HL028 | C1: 40360684 | aa | aa | aa | bb | bb | bb | bb | bb |
| | D105 | C1: 40363439 | aa | h | bb | bb | bb | bb | bb | bb |
| | HL032 | C1: 40373674 | aa | h | bb | bb | bb | bb | bb | bb |
| | HL051 | C1: 41063342 | aa | bb | aa | bb | bb | aa | aa | bb |
| | X12 | C1: 12987 | aa | bb | aa | aa | aa | aa | aa | bb |
| | X29 | C1: 10008626 | aa | bb | bb | bb | bb | bb | bb | bb |
| | X32 | C1: 10010120 | aa | bb | bb | bb | bb | bb | bb | bb |
| | X37 | C1: 10012611 | aa | bb | aa | bb | aa | aa | bb | bb |
| | X300 | C1: 41434284 | aa | h | aa | cc | d | aa | bb | cc |
| | X247 | C1: 42584137 | aa | bb | aa | bb | bb | aa | bb | bb |
| | X501 | C1: 42670038 | aa | bb | aa | aa | bb | aa | bb | bb |
| | X582 | C1: 42772807 | aa | bb | aa | bb | aa | aa | aa | bb |
| | X584 | C1: 42774048 | aa | bb | aa | aa | aa | aa | aa | aa |
| | X585 | C1: 42774887 | aa | h | aa | bb | aa | aa | aa | bb |
| | X590 | C1: 42779091 | aa | h | aa | bb | aa | aa | aa | bb |
| | X594 | C1: 42793815 | aa | h | aa | bb | aa | aa | aa | bb |
| | X538 | C1: 42864334 | aa | cc | bb | bb | aa | cc | aa | aa |
| | X621 | C1: 42873488 | aa | bb | aa | bb | bb | bb | bb | bb |
| | X623 | C1: 42874011 | aa | h | aa | bb | bb | bb | bb | bb |
| | X624 | C1: 42875365 | aa | h | aa | bb | u | aa | u | u |
| | X626 | C1: 42875871 | aa | h | aa | bb | u | cc | u | u |
| | X541 | C1: 42905646 | aa | aa | bb | bb | aa | bb | aa | bb |
| | X516 | C1: 42920689 | aa | bb | bb | bb | bb | bb | bb | bb |
| | X422 | C1: 42975389 | aa | bb | bb | bb | bb | bb | bb | bb |
| | X458 | C1: 43006885 | aa | bb | bb | bb | bb | bb | bb | bb |
| | X461 | C1: 43034544 | aa | aa | aa | bb | bb | aa | bb | bb |
| *BoHL2* | HL202 | C4: 44545337 | aa | aa | bb | bb | bb | bb | aa | aa |
| | HL204 | C4: 44603328 | aa | aa | bb | bb | bb | bb | aa | aa |
| | HL205 | C4:44622468 | aa | aa | bb | bb | bb | aa | aa | aa |
| | HL207 | C4:45681773 | aa | aa | bb | bb | aa | aa | aa | aa |
| | HL208 | C4:45758544 | aa | bb | bb | bb | bb | aa | aa | aa |
| | HL209 | C4:45763718 | aa | bb | bb | bb | bb | aa | aa | aa |
| | HL213 | C4:45955493 | aa | aa | bb | bb | aa | bb | aa | aa |
| | HL223 | C4:46159416 | aa | bb | aa | aa | aa | bb | aa | aa |
| | HL227 | C4:46266526 | aa | h | aa | aa | aa | aa | bb | aa |
| | HL230 | C4:46320472 | aa | aa | bb | bb | bb | aa | aa | aa |
| | HL234 | C4:46336754 | aa | h | bb | bb | bb | aa | aa | aa |
| | HL235 | C4:46408921 | aa | bb | bb | bb | bb | aa | aa | bb |
| | HL239 | C4:47534851 | aa | bb | bb | aa | bb | bb | bb | bb |
| | HL249 | C4:47534851 | aa | h | h | h | aa | aa | aa | aa |
| | HL251 | C4:47619454 | aa | h | bb | bb | aa | bb | aa | aa |
| | L382 | C4:46024699 | aa | bb | bb | bb | aa | bb | bb | aa |
| | L148 | C4:46058658 | aa | bb | bb | bb | bb | aa | bb | aa |
| | L27 | C4:46162407 | aa | aa | bb | bb | bb | aa | aa | aa |
| | L237 | C4:46289641 | aa | aa | bb | bb | bb | aa | aa | aa |
| | L43 | C4:46304511 | aa | aa | bb | bb | bb | aa | aa | aa |
| | L411 | C4:46162346 | aa | aa | bb | bb | bb | aa | aa | aa |
| | L366 | C4:46771685 | aa | cc | bb | bb | aa | cc | aa | aa |
| | L274 | C4:47177899 | aa | cc | bb | bb | aa | aa | aa | cc |
| | L281 | C4:47450452 | aa | cc | bb | bb | aa | aa | aa | cc |
| | L308 | C4:52040075 | aa | h | aa | h | aa | aa | aa | aa |

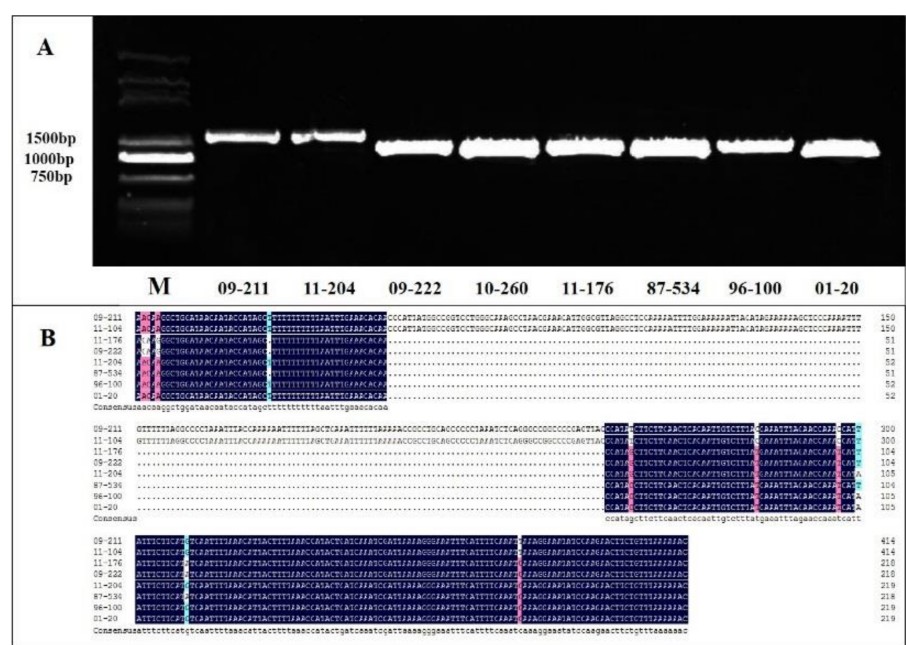

**Figure 2.** Detection of eight cabbage lines by using MBoHL1 and alignment results of Sanger sequencing. (**A**), the amplification products of the *BoHL1* lines were approximately 200 bp longer than the lanes of the other lines; (**B**), the alignment results also suggested that the SV carrying the 195-bp insert was only present in the *BoHL1* lines.

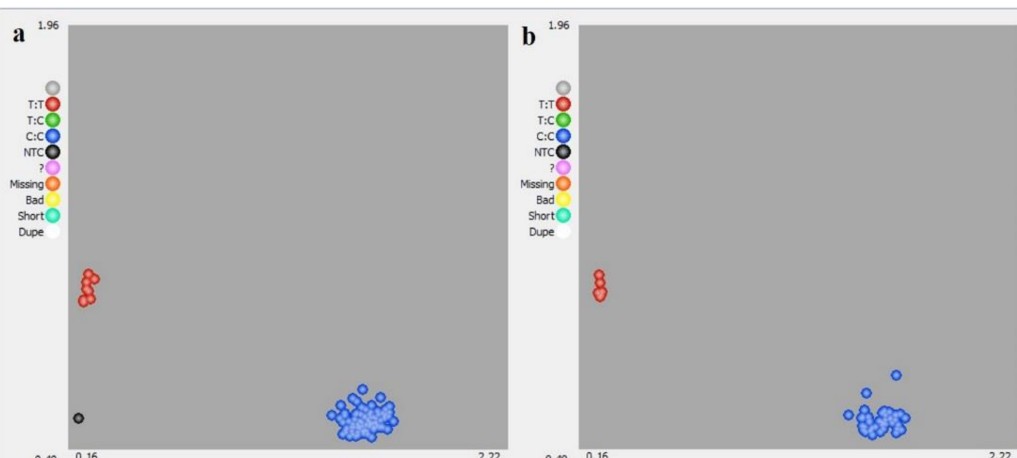

**Figure 3.** Genotype plot for KBoHL2 generated using the KASP assay. T/T (Red dots), genotypic results for 11 *BoHL2* lines; C/C (Blue dots), genotypic results for normal lines; black dots indicate negative controls. (**a**), Genotype plot for 1–100 lines by KBoHL2, (**b**), Genotype plot for 101–138 lines by KBoHL2.

### 3.4. Both MBoHL1 and KBoHL2 Showed 100% Accuracy in Identifying HL Genes in Cabbage

Previously, five hybrid lethal lines, including four cultivated cabbage lines (09-211, 11-204, 09-222 and 11-176) and one wild cabbage line (10-260), were discovered. To identify additional hybrid lethal lines to avoid the HL phenomenon during breeding and further confirm the reliability of our molecular markers, 138 inbred lines with different backgrounds were identified using MBoHL1 and KBoHL2. In the MBoHL1-based analysis, only one cabbage line G94 showed the same bands as the *BoHL1* cabbage lines (Figure 4). In the KBoHL2-based analysis, 11 inbred lines exhibited the A/A allele, which showed the same allele with *BoHL2* lines, including G4, G11, G18, G29, G43, G87, G90, G91, G92, G93 and G95 (Figure 3). Compared with the identification results of the markers HL230, L27, HL209

and L237, two cabbage lines (G73 and G81) were not identified by KBoHL2 as carrying *BoHL2* (Table S4). Above identified HL lines will be further tested using different cross combinations.

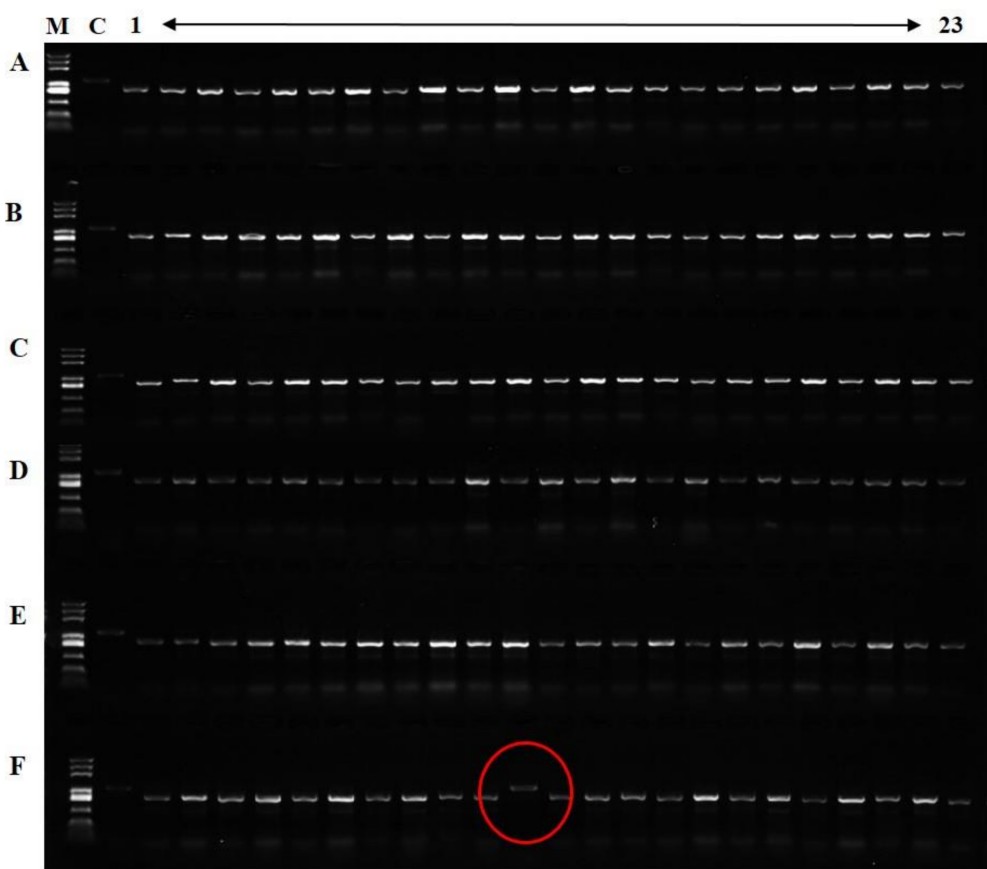

**Figure 4.** Detection of 138 cabbage lines by using MBoHL1. C, positive control, the amplified bands of 09-211. F11 (cabbage line G94) exhibited the same bands as the positive control (The red circle). (**A**) A2-A23, (**B**) B2-B23, (**C**) C2-C23, (**D**) D2-D23, (**E**) E2-E23, (**F**) F2-F23 represent 138 cabbage lines.

Then, 12 inbred lines identified by using MBoHL1 and KBoHL2 and other cabbage lines were crossed 09-211 and 09-222, respectively. The list of different cross combinations is shown in Table S3. All seeds exhibited a 100% germination rate, and in the first developmental stage, they appeared normal. Some cross combinations resulted in the same lethal phenotype observed in the control hybrid lethal $F_1$ (09-211 × 09-222) line, causing 100% seedling mortality (Figure 5). All of the plant lethal genotypes identified by using MBoHL1 and KBoHL2 matched the phenotyping and test crossing results, indicating that the two markers presented 100% accuracy in the rapid identification of hybrid lethal lines. However, the $F_1$ hybrids generated from crosses between G73 and G81 were all normal plants, suggesting that false positives were identified by the previous mapping markers HL230, L27, HL209 and L237. Thus, the *BoHL1*-specific InDel marker MBoHL1 and the *BoHL2*-specific KASP marker KBoHL2 showed 100% accuracy and were reliable markers for the rapid identification of HL genes in cabbage.

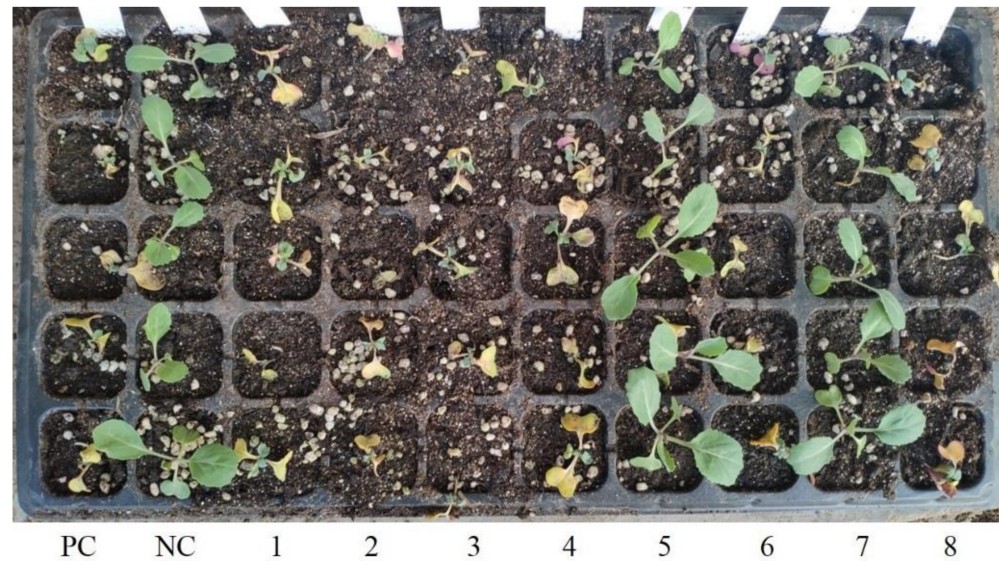

**Figure 5.** Phenotyping and test crossing results. PC, positive control group: hybrid lethality $F_1$ (09-211 × 09-222); NC, negative control group: normal cross combination 09-211 × 87-534; 1, 2, 3, 4, 6 and 8 exhibited the same lethal phenotype with 100% seedling mortality; 5 and 7 appeared normal.

## 4. Discussion

### 4.1. Molecular Marker Technology

Marker-assisted selection is an important tool that is widely used in breeding and enables direct genotypic selection and effective gene polymerization through the application of specific markers associated with target characteristics. In our study, InDel markers associated with *BoHL1* and KASP markers associated with *BoHL2* were successfully developed. First, six fine mapping markers for *BoHL2* showed 100% accuracy in the eight cabbage lines. Compared with the results obtained with KBoHL2, 11 inbred lines carrying *BoHL2* genes were also identified by HL230, L27, HL209 and L237. However, two cabbage lines (G73 and G81) were identified as false positives by HL230, L27, HL209 and L237, suggesting that it is not easy to obtain user-friendly markers from these short InDel markers. InDel markers used for fine mapping should be selected to detect 3-8 bp InDels from a 100–150-bp region, which should make them sensitive to variation within the amplified region. Different background lines may show other variations within an amplified region, reducing its applicability. Moreover, two different kinds of molecular markers show unique advantages in the process of the operation. KASP markers can be used to carry out batch experiments, but such experiments require expensive equipment. InDel markers show high accuracy and stability, helping to avoid confusion in subsequent analyses due to marker specificity and complexity. Thus, it is important to choose appropriate molecular marker types. Recently, KASP markers have been successfully applied in molecular marker studies, such as the codominant KASP markers sunKASP_224 and sunKASP_225, which were developed for marker-assisted pyramiding of Sr26 with other stem rust resistance genes to achieve durable resistance in wheat [34]; M6 was homozygous for six DNA Kompetitive Allele Specific PCR (KASP) markers spanning a 224-kb region linked to Sli in Dutch germplasm; some KASP markers were developed to improve rice eating and cooking quality through marker-assisted selection to cater to the various consumer preferences, especially in Asian areas [35]. In addition, there are other key criteria for obtaining universal user-friendly markers. (1) The generation of accurate data on mutation variations is one precondition. In our study, eight cabbage lines were used as control materials to test these molecular markers along with 20× resequencing data, which provided sufficient data and sufficient inbred lines as a basis for effectively identifying mutations. It is necessary to confirm the accuracy of these mutations using Sanger sequencing because the resequencing data presented false positives (errors for SNP2 were found in the resequencing data). Han et al.

(2019) also confirmed nucleotide variations for developing KASP markers using Sanger sequencing and obtained four 100% polymorphism markers for cabbage male sterility selection [30]. (2) Sufficient control materials with different backgrounds are needed to increase the accuracy and potential applications of the results; in the current study, we adopted a set of 138 different Brassica accessions to meet this criterion. Additionally, to evaluate the application value of the core SNP markers in rice breeding, 481 germplasms were genotyped with three functional KASP markers designed from the sequences of *GBSSI*, *SSIIa*, and *Badh2*, which constitute a convenient and helpful method for excavating elite rice strains for breeding [36]. After further confirmation, InDel markers and a KASP marker were successfully developed for cabbage lines carrying *BoHL1* and *BoHL2*, respectively, which were the first marker set designed to detect HL in cabbage.

### 4.2. Prospect of Preventing Hybrid Lethality

HL hinders gene exchange between different species and limits heterosis exploitation among certain elite parental lines. We have encountered several cases of hybrid lethality in our cabbage breeding work, resulting in direct economic losses. In our study, 12 hybrid lethal cabbage lines were identified by using molecular markers, and this large number exceeded our expectations. Therefore, the development of the user-friendly molecular markers MBoHL1 and KBoHL2 was a necessary step for identifying all cabbage lines carrying HL genes and will help us avoid hybrid lethal combinations in breeding. In addition, further molecular and functional analyses of HL in cabbage will advance our understanding of the molecular mechanisms underlying HL. Additionally, alternative methods might be adopted to prevent HL and avoid extra economic losses, including: (1) Blockage of the key genes or pathways involved in HL. Rubén et al. (2009) reported that salicylic acid (SA) pathway activation is necessary for HL and that overexpression of the salicylate hydroxylase gene (*NahG*) rescues phenotypic cell death [37]. (2) High-temperature environment. High temperature inhibits the death of HL seedlings in cotton [38]. (3) Exogenous hormone treatment. *Nicotiana glutinosa* × *N. repanda* exhibits temperature-sensitive HL with a higher auxin (AUX) content, and exogenous AUX treatment prevents death [39]. (4) γ-ray and ion beam irradiation [40–42].

### 5. Conclusions

In our study, 54 previous mapping markers were closely linked to *BoHL1* or *BoHL2*, and only six markers for *BoHL2* were available when examined in eight cabbage lines (two *BoHL1* lines; three *BoHL2* lines; three lines without *BoHL*); however, they were neither universal nor user-friendly when employed for HL identification in more inbred lines. Then, an InDel in *BoHL1* and an SNP in *BoHL2* were identified, and the corresponding InDel marker MBoHL1 and the KASP marker KBoHL2 were developed and showed 100% accuracy in eight inbred lines. Moreover, one inbred line carrying *BoHL1* and 11 inbred lines carrying *BoHL2* were identified from 138 cabbage inbred lines using the two diagnostic markers. All of the lethal plant genotypes obtained with the two markers matched the phenotype, which was further validated by the crossing results, indicating that the two markers were highly reliable for the rapid identification of HL genes in cabbage.

**Supplementary Materials:** The following are available online at https://www.mdpi.com/article/10.3390/agronomy11050982/s1, Table S1 Nucleotide variations in group 1 in the *BoHL1* region. Table S2 Nucleotide variations in group 2 near the *BoHL2* region. Table S3 List of cross combinations and genotypes. Table S4 Genotype plot for 138 cabbage lines evaluated in this study.

**Author Contributions:** Z.X. performed the experiments and wrote the manuscript. H.L. and Z.F. conceived the study and edited the manuscript. C.K., F.H. analyzed the data and created the figures and tables. L.Y., M.Z., Y.Z., Y.W., J.J. and Z.L. coordinated the study. All authors have read and agreed to the published version of the manuscript.

**Funding:** This study was supported by grants from the National Natural Science Foundation of China (31572139), the Science and Technology Innovation Program of the Chinese Academy of

Agricultural Sciences (CAAS-ASTIP-IVFCAAS) and the earmarked fund for the Modern Agro-Industry Technology Research System, China (CARS-23).

**Institutional Review Board Statement:** Not applicable.

**Informed Consent Statement:** Not applicable.

**Data Availability Statement:** No new data were created or analyzed in this study. Data sharing is not applicable to this article.

**Acknowledgments:** The work was performed at the Key Laboratory of Biology and Genetic Improvement of Horticultural Crops, Ministry of Agriculture, Beijing 100081, China.

**Conflicts of Interest:** The authors declare no conflict of interest.

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
