# Peer review of "Two User-Friendly Molecular Markers Developed for the Identification of Hybrid Lethality Genes in Brassica oleracea"

_agronomy, doi:10.3390/agronomy11050982_

Round 1
Reviewer 1 Report
The manuscript entitled; “Two user-friendly molecular markers developed for the identification of hybrid lethality genes in Brassica oleracea” is an exciting study by Xiao et al where authors developed and validate the two markers’ system to determine the lethality in cabbage hybrids.
Introduction
Line 34-36, please define the heterosis clearly.
Line 36-40, please improve both sentences as English is very weak.
Line 51-53 i.e., Incompatibilities in internal genetic factors also cause effects such as hybrid sterility, hybrid weakness/necrosis, hybrid breakdown, and hybrid inviability” can be improved by “Incompatibilities in internal genetic factors causes hybrid weakness/necrosis, breakdown, sterility and inviability.” Please check and improve throughout the manuscript.
Line 56, please replace the word “and” with “,”.
Line 67, please replace the word “newer” with “recent ones”
Please state the hypothesis of this study in the last paragraph of introduction.
Materials and methods
Line 93, please replace “from” with “by”.
The sentence “and the 122 genotypes were called with Viia 7 software, v1.0.” needs to rewrite.
Results
Line 178, replace “are” with “were”
Line 202-203, Replace “(T/T allele for three BoHL2 lines and C/C allele for five no-BoHL2 lines) (Figure 3)” with “(T/T allele for three BoHL2 lines and C/C allele for five no-BoHL2 lines (Figure 3))
Figure 4, what red circle indicates? please explain in the figure legend.
Discussion
Line 277, please replace “need” with “needed”
Discussion may be improved by giving more examples related to KASP and other markers.
Author Response
Introduction
Line 34-36, please define the heterosis clearly.
Reponse: we have corrected it.
Line 36-40, please improve both sentences as English is very weak.
Reponse: we have corrected it.
Line 51-53 i.e., Incompatibilities in internal genetic factors also cause effects such as hybrid sterility, hybrid weakness/necrosis, hybrid breakdown, and hybrid inviability” can be improved by “Incompatibilities in internal genetic factors causes hybrid weakness/necrosis, breakdown, sterility and inviability.” Please check and improve throughout the manuscript.
Response: we have corrected it
Line 56, please replace the word “and” with “,”.
Response: we have corrected it
Line 67, please replace the word “newer” with “recent ones”
Response: we have corrected it
Please state the hypothesis of this study in the last paragraph of introduction.
Reponse: we have added it
Materials and methods
Line 93, please replace “from” with “by”.
Reponse: we have corrected it
The sentence “and the 122 genotypes were called with Viia 7 software, v1.0.” needs to rewrite.
Reponse: we have corrected it
Results
Line 178, replace “are” with “were”
Reponse: we have corrected it
Line 202-203, Replace “(T/T allele for three BoHL2 lines and C/C allele for five no-BoHL2 lines) (Figure 3)” with “(T/T allele for three BoHL2 lines and C/C allele for five no-BoHL2 lines (Figure 3))
Reponse: we have corrected it
Figure 4, what red circle indicates? please explain in the figure legend.
Reponse: we have added it in the figure legend
Discussion
Line 277, please replace “need” with “needed”
Reponse: we have corrected it
Discussion may be improved by giving more examples related to KASP and other markers.
Reponse: we have added it.
Reviewer 2 Report
In this study, Zhiliang Xiao et al. reported the identification and characterization of two molecular markers for the hybrid lethality genes in Brassica oleracea. This discovery holds promise to improve the breeding process of wild cabbage in practice by preventing the death of hybrids. To develop more accurate and broad markers linked to these death genes, authors performed whole genome DNA sequencing of eight cabbage lines and analyzed the genomic variations linked with BoHL1 and BoHL2. Based on the data, molecular markers are carefully designed and validated. Importantly, breeding studies in the cabbage plant showed that the new molecular markers are highly reliable to rapid identification of the death genes. The experiments are carefully designed and performed to a high standard. The manuscript is well written and ready to the next step for publishment. I only have one minor suggestion that Table 2 can be moved to supplemental files to keep the proper size of main text.
Author Response
Reponse: we have removed Table 2 to the supplemental files Tables S4.
Reviewer 3 Report
Dear author,
I very appreciate your work. I consider it as an interesting and important for the vegetable crops breeding. Based on my observations listed below, the manuscript requires multiple language, stylistic and formal corrections in order to improve it to the perfection. I hope to acceptation of all comments and suggestions with understanding and patience.
In order to improve the readability and clarity use these comments
Many sentences are too much difficult to read, being confusing for both, inappropriate construction of clauses needed to be reformulated or fragmented (lines 31-33, 36-39, 171-183, 208-211, 295-303), and illogical placement to paragraphs (lines 50-51, 221-223).
Using of some words of inappropriate sense in the intended context is also confusing (lines 92, 130, 147, 158, 218, 225, 196).
Informative content of the captions of tables and figures should be revised for low informativeness or incompleteness (Fig 1-3), or for inaccuracy in descriptions (Tab 1, 2)
Moreover, the cited works by Murray and Tompson (1980) and Ramirez-Gonzales et al. (2015) are not presented in the list of references.
These and other revisions and suggestions are clearly visualized and discussed directly in the manuscript pdf in attachments.
Sincerely
Rev.

Author Response
Many sentences are too much difficult to read, being confusing for both, inappropriate construction of clauses needed to be reformulated or fragmented (lines 31-33, 36-39, 171-183, 208-211, 295-303), and illogical placement to paragraphs (lines 50-51, 221-223).
Reponse: we have corrected it
Using of some words of inappropriate sense in the intended context is also confusing (lines 92, 130, 147, 158, 218, 225, 196).
Reponse: we have corrected it
Informative content of the captions of tables and figures should be revised for low informativeness or incompleteness (Fig 1-3), or for inaccuracy in descriptions (Tab 1, 2)
Response: we have added description in figure legendsl
Moreover, the cited works by Murray and Tompson (1980) and Ramirez-Gonzales et al. (2015) are not presented in the list of references.
Reponse: we have corrected it